# OpenReview forum: "AsyncSpade: Efficient Test-Time Scaling with Asynchronous Sparse Decoding"
_ICLR.cc/2026/Conference — Submitted to ICLR 2026_

### Official Review · Reviewer_nRBP · 2025-10-31

**Soundness:** 2
**Presentation:** 2
**Contribution:** 2
**Rating:** 4
**Confidence:** 4

**Summary:**

This paper proposes AsyncSpade, an asynchronous inference framework for sparse decoding of large language models, targeted for test-time scaling scenarios with long reasoning chains. The authors proposed multiple optimization techniques to parallelize the token sparsification and decoding process, yielding an improved throughput at the cost of some drop in task performance.

**Strengths:**

1. The paper is easy to follow.
2. The proposed method effectively parallelizes the token eviction and decoding process, leading to improved throughput.
3. The analysis in Section 3.1 is interesting.

**Weaknesses:**

1. Experimental justifications for some of the core claims are missing.
(1-1.) The authors claim that coarse-grained token selection is suboptimal, and supporting token-level selection is one of the strengths of AsyncSpade. However, they do not provide any experimental evidence or justification for the claim.
(1-2.) One of the claimed core contributions is the next-query prediction algorithm. While the authors provide some indirect analysis in Figure 5, there is no ablation showcasing that the proposed algorithm leads to better task performance (which is what is important at the end of the day) compared to simple alternatives, such as simply using the last available query state without any modification (like TOVA).

2. The efficiency analysis neglects measurements of peak memory requirements, which is another core aspect of inference efficiency.

3. The suggested approach consistently yields a notable performance drop (often dropping over 5% compared to TOVA). It is questionable if the suggested 1.5x~2x speed gain would be worth the performance drop. (Figure 9 suggests a significant gain in terms of FLOPS, but the gain does not translate to a dramatic speedup in Figure 10)

4. There are other high-performing baselines (other than TOVA) that achieve high performance with sparse decoding, such as InfiniPot [1]. While the KV-eviction and decoding happens sequentially, the performance gap might become even larger than TOVA, making the practical applicability questionable.

5. Main experiments only consider models in the Qwen3 family. It is questionable if the findings generalize to other types of models.

[1] Kim et. al., InfiniPot: Infinite Context Processing on Memory-Constrained LLMs, EMNLP 2024

**Questions:**

1. How was the generation budget determined for each model/task in Figure 9?
2. How was the FLOPS measured in Figure 9? Why does such a dramatic reduction in computation lead to relatively modest latency gains in Figure 10?

---

### Official Review · Reviewer_AmvQ · 2025-10-31

**Soundness:** 2
**Presentation:** 3
**Contribution:** 2
**Rating:** 4
**Confidence:** 4

**Summary:**

This paper proposes AsynSpade, a novel asynchronous framework for training-free efficient test-time scaling with a temporal-regressive module and an asynchronous and disaggregated cache architecture.

**Strengths:**

1. This paper aims to address an important question in test-time scaling with sparse attention

2. The results are good and comprehensive.

**Weaknesses:**

1. The main concern I have with the paper is the lack of system evaluation details. It would be better if the authors could provide further implementation details, such as serving backend, parallelism settings, and such.
2. It would be better if the authors could provide further experimental results on scaling across different decoding lengths to showcase the scalability.
3. Some system-related works on KV cache sparsity are missing [1-3].

[1] InfiniGen: Efficient Generative Inference of Large Language Models with Dynamic KV Cache Management, OSDI 2024.

[2] Keyformer: KV Cache Reduction through Key Tokens Selection for Efficient Generative Inference, MLSys 2024.

[3] ALISA: Accelerating Large Language Model Inference via Sparsity-Aware KV Caching, ISCA 2024.

**Questions:**

Please see the weaknesses.

---

### Official Review · Reviewer_m7iG · 2025-11-01

**Soundness:** 3
**Presentation:** 3
**Contribution:** 3
**Rating:** 4
**Confidence:** 3

**Summary:**

The paper introduces a novel framework called AsyncSpade aimed at enhancing the efficiency of sparse decoding in test-time scaling (TTS) tasks for large language models (LLMs).

The paper identifies that the linear growth of key-value (KV) caches during TTS can lead to significant memory constraints. AsyncSpade tackles this issue by implementing a lightweight temporal regression module that predicts the query state for the next token, thereby optimizing memory usage.
AsyncSpade decouples KV cache filtering from the autoregressive decoding loop through an asynchronous operation. This design allows for overlapping token-level KV selection with forward inference computations, effectively eliminating sequential dependencies that can hinder performance.
The framework achieves theoretically optimal TPOT without sacrificing model performance. Experimental results demonstrate that AsyncSpade reduces TPOT by over 20% compared to existing best baselines (Quest) and by at least 50% compared to full attention mechanisms, while maintaining or exceeding accuracy across various TTS benchmarks.
The authors validate their approach through extensive experiments on models such as Qwen3-8B and Qwen3-32B, showcasing the practical benefits of AsyncSpade in real-world applications.

In summary, AsyncSpade represents a significant advancement in optimizing LLM decoding efficiency, particularly for TTS tasks, by addressing memory constraints and improving computational performance without compromising accuracy

**Strengths:**

1. AsyncSpade effectively addresses the memory bottlenecks associated with the linear growth of key-value (KV) caches during test-time scaling (TTS) tasks. By implementing a lightweight temporal regression module to predict query states, it optimizes memory usage, allowing for more efficient processing of large language models (LLMs)
2. The framework decouples KV cache filtering from the autoregressive decoding loop, enabling asynchronous operations. This design allows for overlapping token-level KV selection with forward inference computations, significantly improving the overall efficiency of the decoding process
3. AsyncSpade achieves theoretically optimal TPOT without compromising model performance. Experimental results indicate that it reduces TPOT by over 20% compared to existing best baselines (Quest) and by at least 50% compared to full attention mechanisms, while maintaining or exceeding accuracy across various TTS benchmarks
4. The authors provide extensive experimental validation of AsyncSpade on models such as Qwen3-8B and Qwen3-32B. The results demonstrate the practical benefits of the framework in real-world applications, showcasing its effectiveness in enhancing the performance of LLMs during TTS tasks

**Weaknesses:**

1. While the experiments are conducted on specific models like Qwen3-8B and Qwen3-32B, the paper does not provide extensive validation of AsyncSpade's effectiveness across a broader range of language model architectures. This raises questions about its generalizability and performance consistency in different contexts
2. The framework relies heavily on the assumption that consecutive query states exhibit strong temporal locality and can be accurately predicted using historical data. This dependence may limit its effectiveness in scenarios where the query patterns are less predictable or exhibit significant variability
3. While the paper claims that AsyncSpade reduces the time per output token (TPOT), the introduction of asynchronous operations may lead to overhead in managing communication between different ranks (Inference Rank and Cache Rank). This overhead could negate some of the efficiency gains in certain scenarios, especially under heavy concurrency

**Questions:**

1. How does AsyncSpade perform when applied to a wider variety of large language models beyond Qwen3-8B and Qwen3-32B?
2. What specific conditions or characteristics of query states were observed that support the assumption of strong temporal locality?
3. How do the overheads associated with managing asynchronous operations impact the overall performance in different scenarios, particularly under high concurrency?

---

### Official Review · Reviewer_T3zk · 2025-11-02

**Soundness:** 3
**Presentation:** 4
**Contribution:** 3
**Rating:** 6
**Confidence:** 3

**Summary:**

The proposed AsyncSpade is an asynchronous, disaggregated test-time scaling (TTS) framework that decouples KV-cache selection from the autoregressive decoding loop. It predicts the next-step query embedding from a short sliding window of recent queries via a lightweight ridge-regression scheme, enabling token-level KV selection to run in parallel on a dedicated "Cache Rank” GPU while the “Inference Rank” performs the forward pass.

**Strengths:**

1. The systems insight is clear -- remove the sequential dependency of query-aware sparsity by predicting the next query and running KV filtering asynchronously off the critical path.

2. The method is training-free, simple (ridge regression over recent queries), and broadly compatible with MHA/GQA/MQA variants via batched matmul paths.

3. It is empirically competitive across multiple TTS tasks with consistent TPOT reductions and lower per-token FLOPs; comparisons to Quest/TOVA are comprehensive.

**Weaknesses:**

1. The approach relies on strong temporal locality/linear correlation of adjacent queries; failure modes (e.g., topic shifts, tool-use jumps, abrupt reasoning pivots) are not deeply tested.

2. Results hinge on a two-rank design and sufficient inter-GPU bandwidth. Practicality for single-GPU or consumer GPUs is not fully quantified; fallbacks are unspecified.

**Questions:**

1. How does accuracy/TPOT change when temporal locality breaks (e.g., mid-trajectory domain shift, tool-use, derailments)? Any back-off to page-level selection when prediction error is high?

2. How are Cache/Inference Ranks synchronized to avoid stalls? Provide timeline traces (e.g., Nsight Systems) showing stable overlap across batches and variable sequence lengths.

3. When the predictor misestimates the next query, what is the observed accuracy drop? Any light-weight recheck (e.g., small safety margin tokens) to hedge mispredictions?

---

### Official Review · Reviewer_5mjx · 2025-11-06

**Soundness:** 2
**Presentation:** 3
**Contribution:** 2
**Rating:** 4
**Confidence:** 4

**Summary:**

This paper addresses the computational bottleneck in test-time scaling (TTS) for large language models, where extended chain-of-thought reasoning leads to linear KV-cache growth and memory-bound attention computation. The authors propose AsyncSpade, an algorithm-system co-design that decouples KV-cache management from the inference pipeline through: (1) a temporal-regressive module that predicts next-token query states, and (2) an asynchronous disaggregated architecture with separate Inference and Cache ranks. The method achieves token-level sparsity granularity while claiming to eliminate sequential dependencies. Experiments on Qwen3-8B/32B models show >20% TPOT reduction vs Quest and >50% vs full attention on TTS benchmarks (AIME24/25, GPQA-Diamond, MATH-500).

**Strengths:**

1. **Well-motivated problem**: The paper clearly identifies and quantifies the cache selection bottleneck in existing query-aware sparse attention methods (Figure 2), showing it dominates TPOT under high concurrency/long context scenarios.

2. **Solid empirical observations**: The temporal locality analysis (Section 3) with overlap ratio metrics provides convincing evidence for the prediction approach. Figures 4-5 demonstrate strong linear correlation between adjacent queries (>40% overlap), supporting the prediction strategy.

3. **Novel architectural design**: The disaggregated Inference/Cache rank architecture is creative and well-explained. The asynchronous communication strategy enabling token-level selection while maintaining efficiency is an interesting systems contribution.

4. **Comprehensive hardware testing**: Evaluation on multiple hardware configurations (A100, H100) with detailed latency breakdowns provides good coverage of deployment scenarios.

5. **Token-level granularity**: Unlike prior work (Quest, MoBA) that uses page/block-level selection, AsyncSpade achieves finer-grained token-level selection through the disaggregated architecture.

**Weaknesses:**

1. **Core claim insufficiently validated**: The paper's central innovation is using predicted queries for sparse decoding, but provides no quantitative analysis of how prediction quality impacts final task accuracy. Figure 5 shows overlap ratios 0.56-0.84, but doesn't demonstrate this translates to preserved accuracy.

2. **Missing critical baseline comparisons**:
   - MoBA (Lu et al., 2025) is cited but not experimentally compared despite being recent and relevant
   - ShadowKV (Sun et al., 2025) uses similar temporal locality insights but is not compared
   - TOVA shows anomalous results (131.5ms vs 91.3ms for full attention on H100 32B) with no explanation

3. **Hardware comparison unclear**:
   - AsyncSpade requires "dedicated Cache GPU" (line 118) but paper doesn't clarify if baselines also use multiple GPUs
   - No fair cost/throughput analysis accounting for hardware requirements
   - Different context lengths used across benchmarks without justification (8B: 32k for AIME, 16k for GPQA/MATH; 32B: 32k for AIME, 8k for GPQA/MATH)

4. **Limited reproducibility**: No code provided, many hyperparameters unspecified (exact ε values, packing strategy details), experimental setup lacks details (temperature, sampling method, number of runs/seeds)

**Questions:**

1. **Prediction quality validation** : Can you provide an ablation study showing how prediction quality (measured by overlap ratio or L2 distance) impacts final task accuracy? This is essential to validate your core claim.

2. **Hardware setup clarification** : Does AsyncSpade require 2 physical GPUs? Do baselines use 1 or 2 GPUs? What is the fair throughput/cost comparison accounting for GPU count?

3. **Missing baseline comparisons** : Why were MoBA (Lu et al., 2025) and ShadowKV (Sun et al., 2025) not compared despite being recent and relevant?

4. **TOVA anomaly**: Why does TOVA show 131.5ms on H100 32B when full attention is 91.3ms? This contradicts expectations for a sparse method.

5. **Reproducibility details**: Can you provide: (a) exact ε values used, (b) packing strategy details, (c) experimental setup (temperature, sampling method, number of runs/seeds)?

---

### Meta-Review · Area_Chair_kmza · 2026-01-02

**Summary:**

Reviewers raised the following concerns:
1. Insufficient baseline comparison and experiment details.
2. Limited applications to other models in addition to Qwen 8B and 32B.
3. Missing discussions on other possible overheads introduced by the proposed framework.
4. Missing baseline comparisons.

**Reviewer Concerns:**

The authors didn't provide a rebuttal, so the concerns are still outstanding.

**Reviewer Scores:**

Reviewers will keep their scores because no rebuttal was provided by the authors.

---

### Decision · Program_Chairs · 2026-01-26

Reject